# Genome-wide association studies of COVID-19 vaccine seroconversion and breakthrough outcomes in UK Biobank

Marta Alcalde-Herraiz[1], Martí Català[1], Albert Prats-Uribe[1], Roger Paredes[2,3], JunQing Xie [1,5] & Daniel Prieto-Alhambra [1,4,5] ✉

Understanding the genetic basis of COVID-19 vaccine seroconversion is crucial to study the role of genetics on vaccine effectiveness. In our study, we used UK Biobank data to find the genetic determinants of COVID-19 vaccine-induced seropositivity and breakthrough infections. We conducted four genome-wide association studies among vaccinated participants for COVID-19 vaccine seroconversion and breakthrough susceptibility and severity. Our findings confirmed a link between the *HLA* region and seroconversion after the first and second doses. Additionally, we identified 10 genomic regions associated with breakthrough infection (*SLC6A20, ST6GAL1, MUC16, FUT6, MXI1, MUC4, HMGN2P18-KRTCAP2, NFKBIZ* and *APOC1*), and one with breakthrough severity (*APOE*). No significant evidence of genetic colocalisation was found between those traits. Our study highlights the roles of individual genetic make-up in the varied antibody responses to COVID-19 vaccines and provides insights into the potential mechanisms behind breakthrough infections occurred even after the vaccination.

The emergence of SARS-CoV-2 outbreak presented a major global health challenge, resulting in an unprecedent scale of COVID-19 vaccination intervention globally. Although COVID-19 vaccines showed remarkable effectiveness in preventing severe outcomes and hospitalisation[1,2], they were not equally effective for all individuals due to multifaceted factors. Among all these complex elements, the impact of host genetics on the variability of vaccine-induced seroconversion and breakthrough susceptibility and severity remains unclear.

Large-scale genome-wide association studies (GWAS) have identified over 50 common loci associated with COVID-19 susceptibility and severity[3–13], significantly improving our understanding of the biological mechanisms underlying this complex disease. However, only a few small studies involving trial participants, have shown the genetic variants associated with immune response after vaccination. Understanding the role of genetics on seroconversion and subsequent breakthrough infection and related complications is key to further unravelling the biology of vaccine effectiveness. Additionally, identifying genetic determinants of vaccine response can inform research into personalised vaccination strategies, such as the prioritisation of booster doses for individuals less likely to respond to primary vaccination.

We used UK Biobank data, together with its unique linkage to genetics, serological and public health tests, and health records to perform four genome-wide association studies of vaccine-induced seropositivity, breakthrough infection and severe COVID-19. Furthermore, we performed a colocalisation analysis to study the genetic overlap between the identified traits.

[1]Centre for Statistics in Medicine and NIHR Biomedical Research Centre Oxford, NDORMS, University of Oxford, Oxford, UK. [2]Department of Infectious Diseases and Institut de Recerca de la Sida IrsiCaixa, Hospital Universitari Germans Trias i Pujol, Badalona, Catalonia, Spain. [3]Centre for Global Health and Diseases, Department of Pathology, Case Western Reserve University School of Medicine, Cleveland, OH, USA. [4]Department of Medical Informatics, Erasmus University Medical Centre, Rotterdam, The Netherlands. [5]These authors contributed equally: JunQing Xie, Daniel Prieto-Alhambra. ✉e-mail: daniel.prietoalhambra@ndorms.ox.ac.uk

# Results

## Study cohorts

Our main analyses focussed on white British ancestry subjects (field 22006, genetic ethnic ancestry in UK Biobank). 201,893 participants were included in the SARS-CoV-2 serological antibody study. Among these vaccinated participants, we studied 53,203 within the one dose seroconversion analysis, including 15,046 responders and 38,161 non-responders. Similarly, 42,509 participants were included in the two doses seroconversion analysis: 30,455 responders and 12,054 non-responders (see Fig. 1A and Fig. 2A). The distribution of the number of days between the last vaccination dose and the antibody test date is show in Supplementary Fig. 1.

Among the 398,943 vaccinated UK Biobank participants, 315,620 were studied in the breakthrough susceptibility analysis, including 74,662 SARS-CoV-2 breakthrough infections and 240,661 participants not infected during the study period. Finally, out of the breakthrough susceptibility cases, 3860 were severe COVID-19 infections and 70,802 were considered mild infections for the breakthrough severity analysis (see Fig. 1B and Fig. 2B).

Baseline characteristics of each of the cohorts can be found in Supplementary Table 1. Mean age for all the cohorts was around 66–71, with more presence of females (around 55–58%), and with an index of multiple deprivation around 14–16.

## Genome-wide association (GWAS) analyses

From the 784,256 variants loaded from genotype calls, over 600,000 passed the quality control in all the GWAS and were used to build a whole-genome regression model in REGENIE Step 1. From the 93,095,623 imputed variants, ~9,000,000 passed the quality control and were tested for association with each one of the traits. See Supplementary Fig. 2 for further details.

GWAS identified 13 lead independent variants associated with one-dose seroconversion response. The 13 independent lead variants were distributed among two genomic loci: rs9275109 (CHR = 6; OR = 0.86; $P = 1.0 \times 10^{-26}$) between genes *HLA-DQB1* and *MTCO3P1*; and rs79510369 (CHR = 2; OR = 1.48; $P = 5.3 \times 10^{-10}$) in *PLA2R1* gene (see Table 1 and Fig. 3A). Variants within these genomic loci were found in or near *NOTCH4, BTNL2, HLA-DRA, HLA-DRB9, HLA-DRB5, HLA-DRB6, HLA-DRB1, HLA-DQA1, XXbac-BPG254F23.7, HLA-DOB, TAP2, COL11A2P1,* and *HLA-DPB2* genes (see Supplementary Table 2).

Two-dose seroconversion response was linked to seven lead independent variants distributed among two different genetic loci: rs68033958 (CHR = 6; OR = 0.82; $P = 1.4 \times 10^{-21}$) in *HLA-DQB1* gene, and rs3094055 (CHR = 6; OR = 0.89; $P = 2.1 \times 10^{-10}$) as an upstream of gene *UBQLN1P1* (Table 1, Fig. 3B). Other variants were found in or near *NCR3, UQCRHP1, PRRC2A, BTNL2, HLA-DRA, HLA-DRB1, HLA-DQA1, XXbac-BPG254F23.7* and *HLA-DQB3* (see Supplementary Table 2).

The conducted breakthrough infection GWAS identified 18 lead independent variants associated with post-vaccine SARS-CoV-2 infection susceptibility. Variants were distributed among ten different genomic loci (see Table 1, Fig. 3C): rs73062389 (CHR = 3; OR = 1.22; $P = 4.0 \times 10^{-56}$) in *SLC6A20* gene, rs16861415 (CHR = 3; OR = 0.84; $P = 6.8 \times 10^{-55}$) in *ST6GAL1* gene, rs11673136 (CHR = 19; OR = 1.08; $P = 4.6 \times 10^{-34}$) in *MUC16* gene, rs112313064 (CHR = 19; OR = 1.06; $P = 1.1 \times 10^{-22}$) in *FUT6* gene, rs681343 (CHR = 19; OR = 0.95; $P = 1.4 \times 10^{-17}$) in *FUT2* gene, rs1977829 (CHR = 10; OR = 0.94; $P = 1.4 \times 10^{-12}$) in the *MXI1* gene, rs2550250 (CHR = 3; OR = 1.05; $P = 1.8 \times 10^{-12}$) in *MUC4* gene, rs6676150 (CHR = 1; OR = 1.04; $P = 2.7 \times 10^{-12}$) between *HMGN2P18* and *KRTCAP2* genes, rs17347644 (CHR = 3; OR = 0.96; $P = 1.3 \times 10^{-10}$) in *NFKBIZ* gene, rs5117 (CHR = 19; OR = 0.96; $P = 4.7 \times 10^{-9}$) in *APOC1* gene. Other variants were located in or near *LIMD1, LZTFL1, XCR1, FLT1P1, RPS2OP14, RP1142D20.1,* and *FUT3* (see Supplementary Table 2).

Only one lead variant located in an exonic region of gene *APOE* was associated with breakthrough severity: rs429358 (CHR = 19; OR = 1.21; $P = 1.1 \times 10^{-8}$) (see Table 1, Fig. 3D, Supplementary Table 2).

## Validation analyses

Participants that fulfilled the criteria to be part of the study cohorts but whose genetic ethnic group was not Caucasian (field 22006 in UK Biobank) were included in the validation cohort. In total, we included 8189 (2185 responders, 6020 non-responders) participants in the one-dose seroconversion response cohort, 6533 (4595 responders, 1961 non-responders) participants in the two-dose seroconversion cohort, 57,851 (12,798 infected, 45,437 non-infected) in the breakthrough susceptibility cohort and 12,727 (708 with severe infection, 12,090 with mild infection) in the breakthrough severity cohort. Population characteristics of these cohorts can be found in Supplementary Table 3.

We tested all the lead independent variants for association in the minority ethnic ancestry population. All results can be found in Supplementary Table 4, and results for top lead variants are shown in

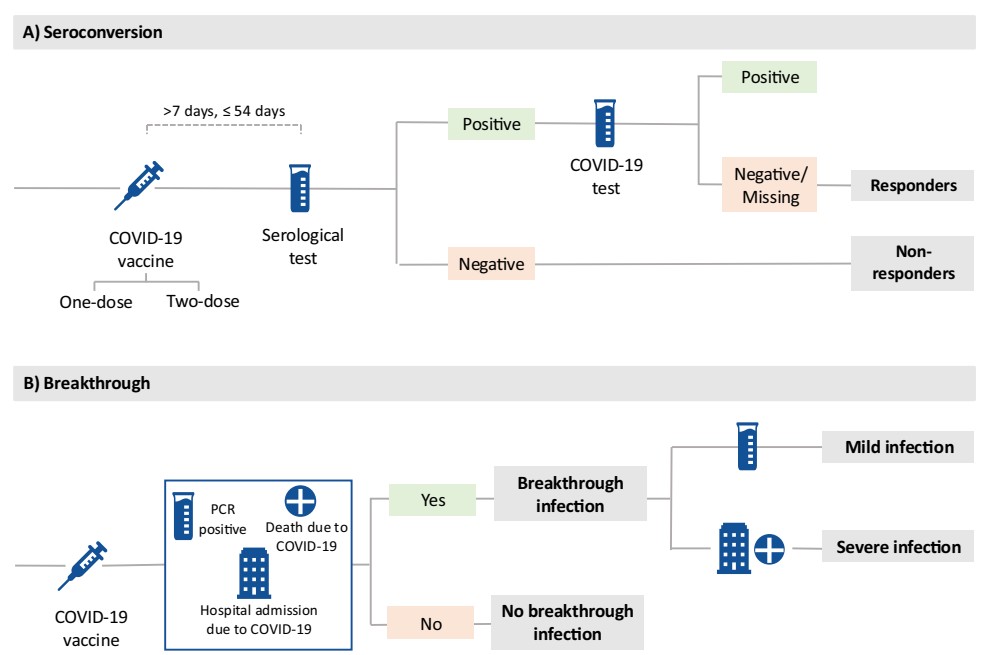

**Fig. 1 | Cases and controls definitions. A** Definitions for seroconversion studies. **B** Definitions for breakthrough studies.

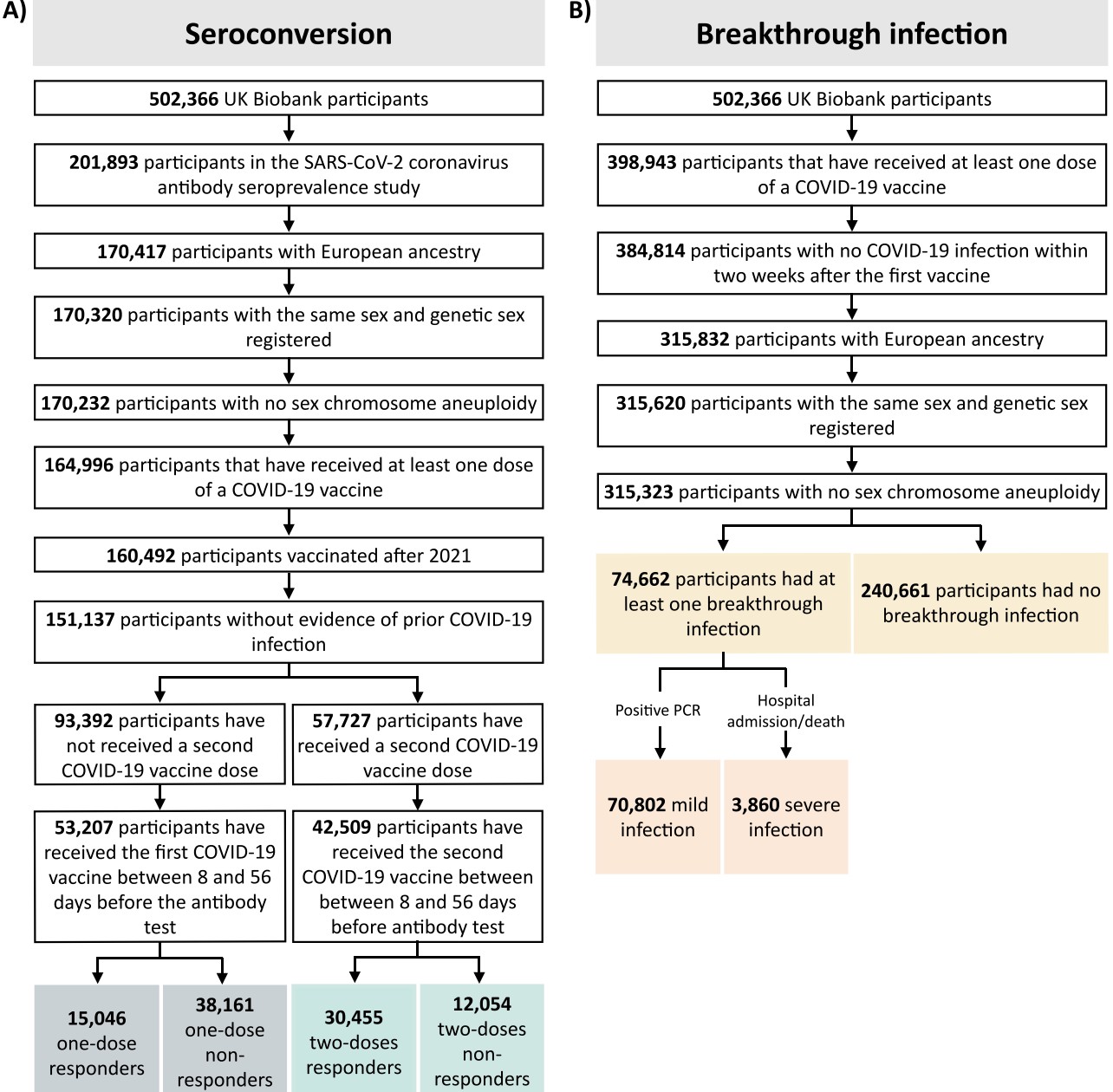

**Fig. 2 | Flow chart for each cohort. A** Seroconversion cohorts, stratified by the number of doses (one or two). Evidence of no prior COVID-19 infection was defined as a negative or missing COVID-19 test result from the COVID-19 infection seroprevalence study. Responders were individuals with a positive serological test from the COVID-19 self-test antibody seroprevalence study. Non-responders were individuals who tested negative in the same study. **B** Breakthrough susceptibility and breakthrough severity cohorts. Breakthrough infection and severity was identified through the linkage to primary care records, hospital inpatient admissions, death registrations, and national infectious diseases surveillance data.

Fig. 4. Regarding top lead variants associated with one dose seroconversion, rs9275109 was validated (OR$_v$ = 0.84; $P_v$ = 1.5·10$^{-5}$), whereas rs79510369 was partially validated (OR$_v$ = 1.10; $P_v$ = 4.5 × 10$^{-1}$). Similarly, one of the top lead variants associated with two-dose seroconversion, rs68033958, was validated (OR$_v$ = 0.86; $P_v$ = 3.3·10$^{-3}$) whereas the other top lead variant, rs3094055, was partially validated (OR$_v$ = 0.98; $P_v$ = 6.7 × 10$^{-1}$).

Additionally, seven of the ten top lead variants associated with breakthrough susceptibility were validated: rs73062389 (OR$_v$ = 1.26; $P_v$ = 4.2 × 10$^{-12}$), rs16861415 (OR$_v$ = 0.89; $P_v$ = 1.0 × 10$^{-4}$), rs11673136 (OR$_v$ = 1.07; $P_v$ = 1.9 × 10$^{-6}$), rs112313064 (OR$_v$ = 1.04, $P_v$ = 5.4 × 10$^{-3}$), rs681343 (OR$_v$ = 0.93; $P_v$ = 1.2 × 10$^{-6}$), rs2550250 (OR$_v$ = 1.06; $P_v$ = 1.8 × 10$^{-4}$) and rs17347644 (OR$_v$ = 0.96; $P_v$ = 2.1 × 10$^{-2}$). The other top

lead variants were partially validated: rs1977829 (OR$_v$ = 0.97; $P_v$ = 1.1 × 10$^{-1}$), rs6676150 (OR$_v$ = 1.02; $P_v$ = 1.1 × 10$^{-1}$) and rs5117 (OR$_v$ = 0.98; $P_v$ = 3.8 × 10$^{-1}$).

The only genetic variant found associated with breakthrough severity in our main analyses was validated in this subpopulation: rs429358 (OR$_v$ = 1.18; $P_v$ = 4.7 × 10$^{-2}$).

**Colocalisation analyses**
We studied the association of all the lead independent variants throughout all the traits (see Supplementary Table 5). We then calculated the probability of a shared causal variant in all the genetic loci. None of the genomic loci colocalised with other traits (see Supplementary Table 6).

**Table 1 | GWAS results of the four different analyses**

| Phenotype | SNP | CHR | BP | EA | OA | EAF | OR | SE | PVAL | Function | Gene/Nearest genes |
|---|---|---|---|---|---|---|---|---|---|---|---|
| Seroconversion - One dose | rs9275109 | 6 | 32,649,676 | T | G | 0.61 | 0.86 | 0.01 | 1.1e-26 | intergenic | HLA-DQB1 - MTCO3P1 |
| Seroconversion - One dose | rs79510369 | 2 | 160,858,048 | T | A | 0.01 | 1.48 | 0.06 | 5.3e-10 | intronic | PLA2R1 |
| Seroconversion - Two dose | rs68033958 | 6 | 32,634,226 | A | G | 0.16 | 0.82 | 0.02 | 1.4e-21 | intronic | HLA-DQB1 |
| Seroconversion - Two dose | rs3094055 | 6 | 30,332,146 | G | C | 0.77 | 0.89 | 0.02 | 2.1e-10 | upstream | UBQLN1P1 |
| Breakthrough susceptibility | rs73062389 | 3 | 45,835,417 | A | G | 0.06 | 1.22 | 0.01 | 4.0e-56 | intronic | SLC6A20 |
| Breakthrough susceptibility | rs16861415 | 3 | 186,696,364 | C | T | 0.08 | 0.84 | 0.01 | 6.8e-55 | intronic | ST6GAL1 |
| Breakthrough susceptibility | rs11673136 | 19 | 9,007,748 | G | A | 0.48 | 1.08 | 0.01 | 4.6e-34 | intronic | MUC16 |
| Breakthrough susceptibility | rs112313064 | 19 | 5,831,724 | C | T | 0.36 | 1.06 | 0.01 | 1.1e-22 | exonic | FUT6 |
| Breakthrough susceptibility | rs681343 | 19 | 49,206,462 | T | C | 0.50 | 0.95 | 0.01 | 1.4e-17 | exonic | FUT2 |
| Breakthrough susceptibility | rs1977829 | 10 | 111,975,041 | A | G | 0.18 | 0.94 | 0.01 | 1.4e-12 | intronic | MXI1 |
| Breakthrough susceptibility | rs2550250 | 3 | 195,500,549 | T | C | 0.44 | 1.05 | 0.01 | 1.8e-12 | intronic | MUC4 |
| Breakthrough susceptibility | rs6676150 | 1 | 155,123,837 | C | G | 0.39 | 1.04 | 0.01 | 2.7e-12 | intergenic | HMGN2P18 - KRTCAP2 |
| Breakthrough susceptibility | rs17347644 | 3 | 101,547,733 | T | C | 0.35 | 0.96 | 0.01 | 1.3e-10 | intronic | NFKBIZ |
| Breakthrough susceptibility | rs5117 | 19 | 45,418,790 | C | T | 0.24 | 0.96 | 0.01 | 4.7e-09 | intronic | APOC1 |
| Breakthrough severity | rs429358 | 19 | 45,411,941 | C | T | 0.15 | 1.21 | 0.03 | 1.1e-08 | exonic | APOE |

Top lead independent variants (P-value ≤ 5 × 10⁻⁸, $r^2$ ≤ 0.1 and window = 250 kb) in FUMA are noted. Association was tested by logistic regression in REGENIE.
SNP single nucleotide polymorphism, CHR chromosome, BP Base pair, EAF effect allele frequency, EA effect allele, OA other allele, OR odds ratio, SE standard error, PVAL P value

## Discussion

To our knowledge, this is the largest GWAS study of COVID-19 vaccine seroconversion response and vaccine effectiveness against infection and severe disease. Our study found variants in the *HLA* region associated with both first dose and second dose vaccine-induced seropositivity. Furthermore, the two top lead variants located in this region (rs9275109 for one dose seroconversion and rs68033958 for two dose seroconversion) were validated in the mixed ethnic cohort. The other independent variants distributed among these loci were located in/near genes such as *UBQLN1P, NOTCH4, BTNL2, XXbac-BPG254F23.7, TAP2, COL11A2P1, NCR3, UQCRHP1, PRRC2A, BTNL2,* and *XXbac-BPG254F23.7.*

Human leucocyte antigens (*HLA*) have been previously acknowledged as the most influential genetic factors for seroconversion to various vaccines[14–16]. Mentzer et al.[17] recently investigated genetic variations associated with 28-day COVID-19 one-dose vaccination antibodies in a cohort of 1076 participants enroled in ChAdOx1 nCov-19 vaccine efficacy trials. Their study pinpointed the association between *HLA* locus with IgG antibody levels and risk of breakthrough infection. Although our recent study[18] showed that the effect size of those genetic associations discovered from highly selective trial's participants may not be fully generalisable to the wider community population, our findings remain supportive that *HLA* region plays a key role in antibody response to COVID-19 vaccines. We also found a signal within the *PLA2R1* (rs79510369), but it was not fully validated in the mixed ancestry cohort.

We further discovered ten genomic loci (*SLC6A20, ST6GAL1, MUC16, FUT6, FUT2, MXI1, MUC4, HMGN2P18-KRTCAP2, NFKBIZ,* and *APOC1*) linked to breakthrough infection and one (*APOE*) associated with breakthrough COVID-19 severity. *SLC6A20, ST6GAL1, MUC16, FUT6, FUT2, MUC4, NFKBIZ* and *APOE* were validated in the mixed ethnic group, whereas the other loci (*MXI1, HMGN2P18-KRTCAP2* and *APOC1*) were partially validated.

More than 50 variants have been associated with COVID-19 susceptibility and severity among unvaccinated subjects in previous studies[3–13]. *SLC6A20* and *FUT2* have been already related to COVID-19 susceptibility and severity in previous GWAS (see Supplementary Table 7). In our study, both were confirmed to also be associated with the risk of COVID-19 infection even after the vaccination. SLC6A20 encodes the sodium-imino acid (proline) transporter 1, also known as *SIT1*. *SIT1* has an important role interacting with angiotensin-converting enzyme 2 (*ACE2*), which is the receptor found in SARS-CoV-2 virus infecting cells. Therefore, evidence suggests that the interaction between *SIT1* and *ACE2* might influence how the virus infects cells[7,19,20]. On the other hand, *FUT2* gene is responsible for secretor status of ABO antigens, which has been proven to influence the susceptibility to some infectious diseases[21]. Taken together, these genetic associations, independent of vaccination status, suggest that viral entry or replication could be more likely to be the potential pathogenesis involved with COVID-19 susceptibility than immune responses, and can inform the focus of future therapeutic targets in the post-pandemic era.

*MUC4* and *MUC16* (genes associated with mucosal immunity) have been previously linked to COVID-19 severity. In a study including 125 hospital-admitted COVID-19 infected participants[22], these genes were observed to be upregulated within the recovered participants, suggesting an active defence against SARS-CoV-2 infection. Similarly, *NFKBIZ* has not been previously associated with COVID-19 susceptibility but reported to have an association with the risk of developing severe COVID-19[23].

Our study found genetic loci associated with breakthrough COVID-19 outcomes (*ST6AL1, FUT6, MXI1, HMGN2P18-KRTCAP2, APOC1*) that have not been previously reported. Although further investigation is required to illuminate specific mechanisms by which these genes operate, they seem to be associated with glucose metabolism pathways. *ST6GAL1* is

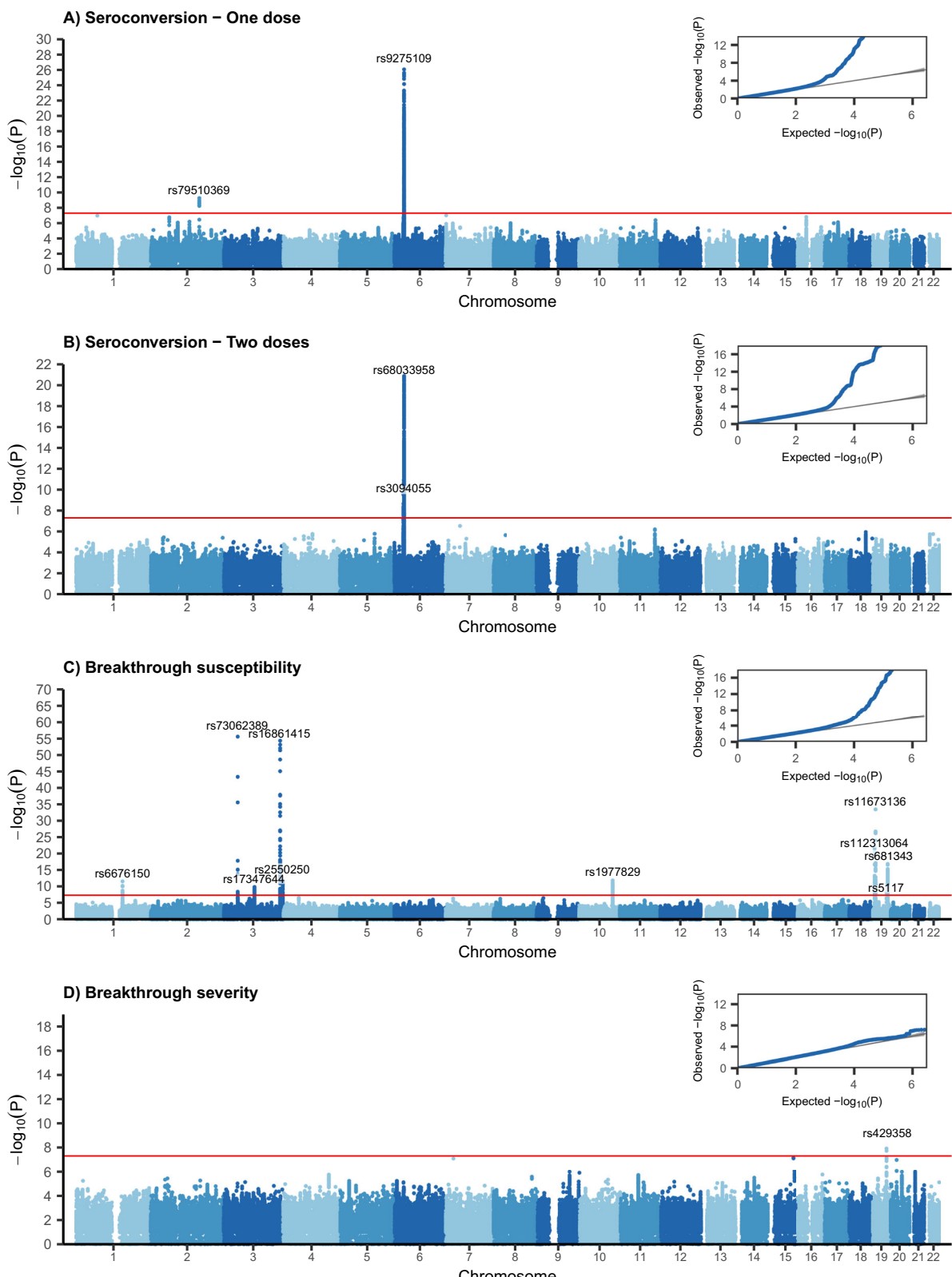

**Fig. 3 | Manhattan plots and QQ Plots for each one of the GWAS.** Notice that only top lead variants of each genetic loci are identified. The *P* Value of the Wald test in (**A**–**D**) is presented raw and did not correct for multiple testing. Source data is provided for this figure. **A** One dose seroconversion results. **B** Two doses seroconversion results. **C** Breakthrough susceptibility results. **D** Breakthrough severity results.

linked to sialic acid, which has been studied as a potential target for the dissemination of SARS-CoV-2 virus[24,25].

Our study also suggested an association between breakthrough severity and *APOE* locus. *APOE* gene is well-known for its association with Alzheimer's disease (AD)[26,27]. Previous evidence suggests that people with AD exhibit elevated morbidity and mortality of COVID-19[28].Therefore, we propose that the *APOE* gene may influence breakthrough severity through its link to Alzheimer's disease.

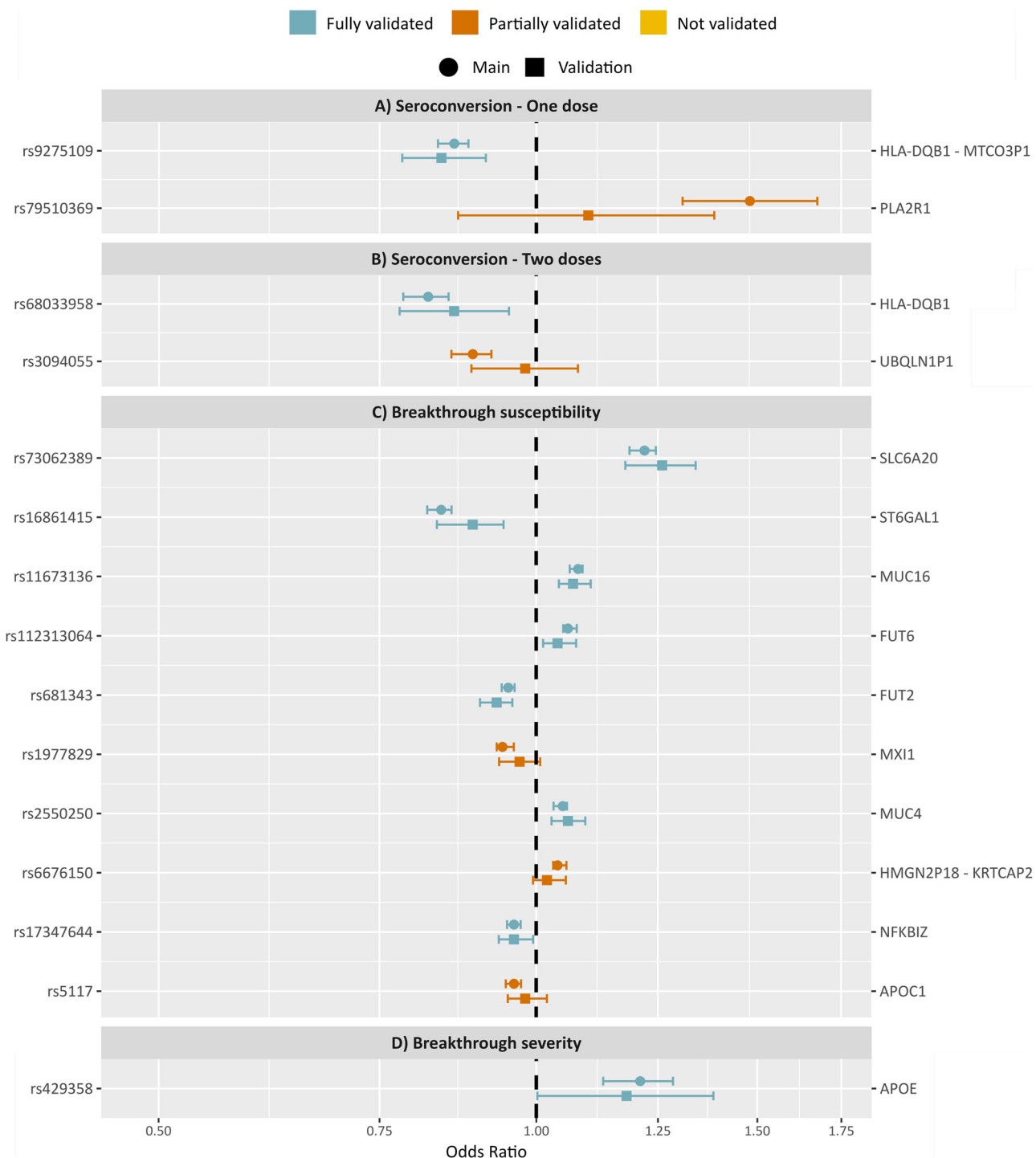

**Fig. 4 | Validation results.** Data are presented as OR ± 95% confidence interval. SNPs in blue are those fully validated, meaning that the OR from the main analysis and the OR from the validation analysis have the same direction, and that the validation *P* value ≤ 0.05. SNPs in orange are partially validated, indicating that both OR have the same direction, but the validation *P* Value > 0.05. SNPs in yellow were not validated, showing that both OR have opposite directions. We used the REGENIE method to perform the statistical test. Source data is provided for this figure. **A** One dose seroconversion results, with a sample size of 8189. **B** Two doses seroconversion results, sample size of 6533. **C** Breakthrough susceptibility results, sample size of 57,851. **D** Breakthrough severity results, sample size of 12,727.

Our study has limitations. First, we only accounted for a binary seroconversion response, which may have reduced GWAS statistical power compared to quantitative antibody levels. However, this is due to the use of a rapid validated lateral flow device, which allowed for the collection of the largest cohort of individuals self-tested for vaccine immune/antibody response to date, including over 200,000 participants. Second, while seropositivity rates in our study are lower than those reported in clinical trials, this can be attributed to the relatively high age of our participants and the reliance on a single serological test result within a time window, which may not capture all individuals that seroconvert. Thirdly, we used linked routinely collected data to ascertain status of COVID-19 infections. This may lead to the outcome misclassification given that asymptomatic or mild infection cases are common particularly among the vaccinated individuals who may not seek for testing and thus falsely classified as controls. Forth, our study population is relatively old (66–71 years old), so care must be taken

when generalising the results to a wider population. Fifth, we did not differentiate between different types of COVID-19 vaccines, which may have distinct breakthrough infection mechanisms. Sixth, external replication of the genetic association in other populations is currently impossible in our study due to the lack of similar linked data elsewhere. To mitigate this, we conducted an internal validation by splitting our cohort into those with European and non-European ancestry. Although we did not find apparent evidence of ethic-specific genetic effects, further studies with larger and more diverse population remain needed to guarantee the findings. Lastly, sample size in genome-wide association studies plays a key role in detecting signals. Some of our analyses, especially the breakthrough severity phenotype, might be underpowered, making it more challenging to detect significant variants.

Despite all these limitations, our study has several strengths that enhance the reliability of our findings. First, we used the largest sample size to date to perform the genome-wide association study. Second, most of the variants found to be associated with the traits in a European population were also found to be associated in a non-European population, enhancing the robustness of our findings. Third, we provide findings about the genetic mechanisms of vaccine seroconversion and breakthrough susceptibility and severity, which may provide valuable insights for the research into personalised vaccination strategies.

## Methods

### Study design and participants

**UK Biobank.** Study participants were from the UK Biobank study, a population-based cohort that recruited over 500,000 participants from England (89%), Wales (7%), and Scotland (4%). All people in the National Health Service registry who were aged 40–69 years and lived <25 miles from a study centre were invited to participate between 2006 and 2010. 503,325 participants were recruited from 9.2 million mailed invitations. Its study design and participant characteristics have been previously described in detail elsewhere[29].

The UK Biobank includes information on demographics, socio-economics, lifestyle factors, physical metrics, and medical history. It also includes genotyping data[30], and follow-up data through linkages to electronic health record databases.

**COVID-19 self-test antibody seroprevalence study.** Alive UKB participants were invited to participate in a SARS-CoV-2 coronavirus antibody seroprevalence study from February 2021 to July 2021. At the same time, UK's vaccination programme was being implemented. All participants, regardless of the sex, age, and health status, that met the inclusion criteria were eligible for recruitment and consequently, received an email with the invitation and brief information about the study. Participants unable or unwilling to participate were encouraged to inform about their decision. After seven days of the original invitation, a remainder email was sent to participants who had not responded yet. Participants willing to participate were asked to confirm their contact details and to give consent to receive a lateral flow self-test kit at their home address. Participants addresses were securely transferred to a third-party mailing house and shipping company, and three days prior to kit dispatch participants were sent an email to let them know that their kit was being dispatched.

Once participants have performed the test, they were asked to complete an online UK Biobank questionnaire. The questionnaire collected information about their result (IgM or IgG positive, negative, invalid), test date, their COVID-19 first and second vaccination status, and dates. Date on which participants submitted their results was automatically recorded. A reminder email was sent to participants who had not returned their test result one week after the kit was dispatched.

Recruitment was in two phases. The first phase invited ~34,713, ~22,390, and ~21,405 participants sequentially. The second phase invited the remaining ~371,985 participants who were not eligible for inclusion in phase one. More details about the study design can be found in the online document: https://biobank.ndph.ox.ac.uk/showcase/label.cgi?id=998.

**COVID-19 infection seroprevalence study.** The lateral flow test device used in the SARS-CoV-2 coronavirus antibody seroprevalence study could not distinguish between antibodies induced by infection or by vaccination. Hence, individuals who had previously participated in the self-test antibody study, who had reported a positive test result, and who had reported being vaccinated prior to taking the antibody test, were re-invited to provide a sample of capillary blood to test for IgG antibodies to the nucleocapsid (N) protein, which is an indicative of a past COVID-19 infection. Recruitment of participants for the COVID-19 infection study was similar to that of the antibody study and more details are provided in the online document: https://biobank.ndph.ox.ac.uk/showcase/label.cgi?id=997.

**Data linkages.** UK Biobank follow-up of the participants is conducted through individual-level linkages to multiple electronic health databases. The databases used in this study include primary care records (prescriptions and diagnoses), hospital inpatient admissions (diagnoses), death registrations, and national infectious diseases surveillance data (COVID-19 test results)[31].

**Genotyping and imputation.** Genotyping and quality control of the genetic dataset of UK Biobank has been described previously[30]. In summary, UK Biobank genotype calling was performed by Affymetrix and includes 784,256 autosomal variants. Imputation was done by combining two different reference panels, the Haplotype Reference Consortium (HRC) and the UK10K haplotype resource. It includes 93,095,623 autosomal SNPs.

### Definition of the study cohorts

In this study, we analysed four different cohorts to study the genetic variants associated with four different traits: (1) Seroconversion induced by one dose of COVID-19 vaccine, (2) seroconversion induced by two doses of COVID-19 vaccine, (3) breakthrough infection susceptibility, and (4) breakthrough infection severity.

In the main analysis, individuals with no European genetic ancestry (field 22006 in UK Biobank), with sex chromosome aneuploidy, and with different sex and genetic sex registered, were not included in the main analysis to avoid confounding effects.

### Seroconversion responders and non-responders

For the seroconversion groups (one dose, two doses), we first restricted the analysis to vaccinated UK Biobank participants that enroled the SARS-CoV-2 seroprevalence study.

Responders were defined as participants with a positive serological test (from the COVID-19 self-test antibody seroprevalence study) within 8 to 56 days after latest dose of vaccination. However, individuals that after a seropositivity result had evidence of prior COVID-19 infection (positive result from the COVID-19 infection seroprevalence study), were excluded from the cohort. Non-responders were participants with a negative serological test within 8–56 days post-vaccination (see Fig. 1A).

### Breakthrough susceptibility and severity

For the breakthrough groups (breakthrough susceptibility, breakthrough severity) we included UK Biobank participants that have received at least one dose of a COVID-19 vaccine.

Breakthrough susceptibility infections were defined by a positive PCR test, hospital admission with a COVID-19 diagnoses (ICD-10 Codes: U07.1, U07.2), or a death certificate listing COVID-19 as the cause of death (same ICD-10 codes). Absence of breakthrough infection were participants with no positive PCR test, no hospital admission with a COVID-19 diagnoses, neither a death certificate listing COVID-19 as the

cause of death. Among breakthrough infection cases, we defined severe COVID-19 cases as those requiring hospitalisation or resulting in death. Mild infections for breakthrough severity analysis were participants with only a PCR test (see Fig. 1B). More details of the ICD-10 codes can be found in Supplementary Note 1.

## Genome-wide association study

We conducted four different genome-wide association studies: one-dose seroconversion, two-dose seroconversion, breakthrough susceptibility and breakthrough infection. Associations between variants and traits were calculated using REGENIE (version 1.0.5)[32], a machine-learning method. Briefly, REGENIE fits a whole genome regression model in two main steps. In step 1, a whole genome regression model is fit using a subset of the total set of available genetic markers. In step 2, a larger set of markers are tested for association conditional upon the prediction from the regression model in step 1 with the trait of interest.

In our study, we used UK Biobank genotype calls for the first step and UK Biobank imputed data in the second step. Regression models were adjusted for baseline age (at the date of antibody testing), sex, genetic batch, and the first ten genetic principal components. We also applied first-correction for imbalanced cases. Variants from both datasets (genotype calls and imputed data) underwent quality control before being used for the analysis. The quality control was performed using PLINK2 (version 1.0.6)[33]. Excluded variants include those with missing genotype data, that deviate from Hardy-Weinberg equilibrium ($P < 1 \times 10^{-15}$), and those with a minor allele frequency <1%. For imputed variants, we also removed duplicated SNPs, keeping only the first instance.

After performing the genome-wide association study, variants with a $p$-value $\leq 5 \times 10^{-8}$ were considered to have a significant statistical association with the traits. Effect sizes of genetic association for each SNP were measured using odds ratios (OR). We used FUMA (version 1.6.0)[34] to identify lead independent significant SNPs ($r \leq 0.1$) and the top lead independent significant SNPs from each genetic locus (window 250 kb). We used positional mapping (window 10 kb) to map SNPs to genes.

Data curation for the generation of analytical cohorts and outcomes was done using R software (Version 4.3.0). GWAS were performed on UK Biobank RAP platform. Plots were generated with R.

## Validation

To study if the obtained associations in the European population were reproduced in other ancestral groups, we tested the independent variants for association using a non-European population. For this analysis, we employed the complete genotype calls data for the 1st step of the REGENIE method. Subsequently, in the 2nd step, we exclusively tested the lead independent significant SNPs obtained in the main analysis.

Variants with the OR from the main analysis and the validated OR pointing to the same direction (OR and $OR_v$, both being >1 or <1) and with a validated $p$-value ($P_v$) $\leq 0.05$ were fully validated. Variants with both OR having the same direction but with a validated $p$-value > 0.05 were partially validated. Variants with OR in opposite directions were not validated.

## Colocalisation

Once we have obtained the variants associated with one dose vaccine seroconversion, two dose vaccine seroconversion, breakthrough infection, and breakthrough severity, we aimed to study the genetic overlap between the different traits. Firstly, we studied the association of all the lead independent variants throughout all the traits. Afterwards, we used the R package coloc[35] to perform a colocalisation analysis within ±250 kb from each genetic locus. Colocalisation analysis enables the assessment of whether the same traits have similar genetic roots. We employed the default prior probabilities for colocalisation: $P_1 = 10^{-4}$, $P_2 = 10^{-4}$, and $P_{12} = 10^{-5}$ and considered a posterior probability for a shared common causal variant assumption (H4) greater than 50% as substantial evidence of colocalisation.

## Software/implementation

We used UK Biobank RAP platform, specifically the software REGENIE[32] (version 1.0.5) and PLINK2[33] (version 1.0.6), to perform the GWAS. We used FUMA[34] (version 1.6.0) to detect independent SNPs. Data manipulation was done with R software (version 4.3.0), and the main packages used were coloc[36] (version 5.2.3), dplyr[37] (version 1.1.3), and ggplot2 (version 3.5.1).

## Reporting summary

Further information on research design is available in the Nature Portfolio Reporting Summary linked to this article.

## Data availability

UK Biobank individual level data can be accessed by applying for access at http://ukbiobank.ac.uk/register-apply/. Ethics approval for the UK Biobank was granted by the North West Multi-Centre Research Ethics Committee in 2006 and was updated regularly after that (https://www.ukbiobank.ac.uk/learn-more-about-uk-biobank/about-us/ethics). All participants provided informed written consent to take part in the study and be followed-up through linkage to health-related records. This study received ethical approval from the UKBB Ethics Advisory Committee (EAC) under application 98358. Source data are provided in this paper. GWAS results are publicly available at the GWAS Catalogue under study accession codes GCST90448693, GCST90448694, GCST90448695 and GCST90448696. Source data are provided with this paper.

## Code availability

All the analytic code is publicly available https://github.com/oxford-pharmacoepi/GeneticDeterminantsCovid19Vaxs[38].

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

## Acknowledgements

D.P.A. receives funding from the UK National Institute for Health and Care Research (NIHR) in the form of a senior research fellowship. DPA's group received partial support from the Oxford NIHR Biomedical Research Centre. J.Q.X. is funded through Jardine-Oxford Graduate Scholarship and a titular Oxford Clarendon Fund Scholarship. The authors express their sincere gratitude to all UK Biobank participants for generously providing an invaluable resource to advance scientific research.

## Author contributions

Conceptualisation (M.A.H., J.Q.X., D.P.A.); data curation (M.A.H.); statistical analysis (M.A.H.); supervision (J.Q.X., D.P.A., M.C., A.P.U., R.P.); interpretation of data (M.A.H., J.Q.X., D.P.A., R.P.); draughting of the manuscript (M.A.H., J.Q.X.); critical revision of the manuscript (J.Q.X., D.P.A., M.C., A.P.U., R.P.). All authors reviewed and approved the final version. The views expressed in this article are the personal views of the author(s) and may not be understood or quoted as being made on behalf of or reflecting the position of the regulatory agency/agencies or organisations with which the author(s) is/are employed/affiliated.

## Competing interests

D.P.A.'s department has received grant/s from Amgen, Chiesi-Taylor, Lilly, Janssen, Novartis, and UCB Biopharma. His research group has received consultancy fees from Astra Zeneca and UCB Biopharma. Amgen, Astellas, Janssen, Synapse Management Partners and UCB Biopharma have funded or supported training programmes organised by DPA's department. R.P. has participated in advisory boards for Pfizer, Gilead, MSD, GSK, Atea, Lilly, Roche, Astra-Zeneca, ViiV Healthcare and Theratechnologies, has participated in lectures and seminars funded by Gilead, Pfizer, GSK and AstraZeneca, and has received research funds awarded to his institution from Gilead, Pfizer, and MSD. All other authors declare no conflicts of interest.
