## [Peer Review File · Nature Communications]

Genetic determinants of COVID-19 vaccine seroconversion and breakthrough infection risk using UK BiobankEditorial Note: Parts of this Peer Review File have been redacted as indicated to remove third-party material where no permission to publish could be obtained.

REVIEWER COMMENTS

Reviewer #1 (Remarks to the Author):

The publication of these data and results would be timely and of considerable interest. What the study suffers in messiness (or more formally "noise") it makes up for with sample size.

Some of the main limitations of the study are acknowledged and discussed, especially the fact that most of the antibody response data are non-quantitative and the result of self-administered home tests (if I understand correctly). The sensitivity and specificity of these tests are probably additional concerns (worsened by variable comprehension and care of the self-administered instructions), and the biggest limitation of all may be the variable time between receipt of a vaccine dose and the administration of the test. Presumably a large number of tests were performed within mere days of the vaccination, which may contribute to high apparent non-response? For the second dose in particular (if not also the first dose), the response rate strikes me as low, but maybe the authors can compare these with more careful and quantitative vaccine trial data. They acknowledge that vaccine type is not accounted for, and may be a significant source of heterogeneity. There is a credible effort at distinguishing between positive antibody test results from prior infection vs vaccine alone, although I didn't see these data presented and actually used fully. The statistical power for the severity tests and perhaps even susceptibility are much reduced and is probably an unacknowledged reason for the diminishing number of associations, as well as the crudeness of the definition of severity and the likelihood that large numbers of mild cases went unreported and unrecognized by the subjects. The fact that HLA class II alone seems to be driving antibody response (to a surface antigen) seems to be a common hallmark of vaccine response studies of this sort. So it's a nice result and adds more data to the edifice on this subject. In many ways Figure 2C may be the most interesting, since this turns up additional associations which probably relate to the antigenically more complicated response to a full viral infection. One would predict these associations would more closely resemble the COVID susceptibility/severity studies prior to or independent vaccination and I think these results could be more explicitly and fully compared with these prior GWAS's. The question of how fully a single (surface-)antigen vaccine can protect vs. more antigenically complicated vaccination strategies seems a worthwhile avenue to explore in light of results like these.

I suppose the use of REGENIE for the association analysis is fine, although I was not familiar with it previously and it's not clear it offers any tangible advantages here over the traditional GWAS analytical approach. It seems to have been developed for the simultaneous analysis of multiple independent traits on large biobank style GWAS data sets, which probably isn't much of an issue here. Also the two stage design of directly typed followed by imputed SNPs, while probably fine, didn't seem to add very much. Were any associations visible only with imputed variants? If not, they seem to have added nothing here, although presumably they could aid with fine-mapping or identification of more directly functional variants. But none of that was shown or discussed here.

Also, the validation of the results in non-European descent individuals is worthwhile but potentially problematic. It's unclear who these people are, and whether they are simply all non-whites from

throughout the world or a single large sub-population within Britain (e.g., South Asian or African/Carribbean). If it's simply all non-whites that is potentially problematic. And of course, if these data are meaningful at all, the fairly large number of partially validated results in Fig. 3 demand more full exploration. And the two that show opposite associations (or total non-validation) for the 2-dose vaccine response may say something important. It may also indict the decision to consider one vs. two dose responses. Incidentally, is 1st dose response data also available for the 2-dose response group, since it appears individuals were only included in one of the two analyses. If 1st dose data are available it might be worth including them in a 1st dose analysis, even if this complicates the accounting for non-independent data in the 2nd dose study.

Reviewer #2 (Remarks to the Author):

This manuscript documents a GWAS in a large cohort (200,000+) of individuals who received COVID-19 vaccination. The outcomes of interest including: serologic response to vaccination, breakthrough infection, and development severe disease. The manuscript is well-written, covers an important topic, has results that nicely complement the multiple publications looking at genetic influences on COVID-19 disease susceptibility and/or severity. The data support the conclusions drawn. The methods are clearly documented and reflect standard approaches typically used.

There are a number of items that the authors should address to strengthen the manuscript and provide increased clarity for readers.

The title and abstract mention antibody response, leading to the assumption that vaccine immunogenicity and magnitude of response will be evaluated. This is not correct, the primary analysis outcome is actually lateral flow assay results that provide a yes/no indicator of seroconversion. In the current manuscript this is not clarified until the limitations paragraph in the Discussion (line 161). This distinction is important and needs to be made clear early on in the manuscript. The authors could change the title from 'antibody response' to 'seroconversion' and/or spell out the outcome in the abstract (line 38-39).

Following up on this – the authors don't have the actual results of the assay, but are relying on self-reported results. This should be more clear in the limitations paragraph in the Discussion. Also, the antibody test was taken within 84 days of vaccination. Why this time period? 84 days provides plenty of time for antibody waning. This needs to be addressed – the analysis may not separate out responders from true non-responders, the latter group may include low responders whose titers waned below the level of detection by the time they tested.

Another issue is the repeated use of 'cases' and 'controls'. Each of the 4 analyses use these terms but define them differently. It would be more clear if the authors could use different terminology for each analysis. For example: one-dose responders vs non-responders; two-dose responders vs non-responders; breakthrough infection vs no infection; mild disease vs severe disease.

No details are provided for the validation cohort of non-European ancestry. Nor was there any

discussion of how the different genetic makeup affects the validation results. What about genetic variants that operate in Caucasians but not other racial groups (or vice versa)?

The discussion section lacks details about the genes associated with each outcome. It does provide information from other studies that have found the same or similar regions, but there is no information given about what each gene does or how it might be related to antibody response, breakthrough infection, or disease severity. The one exception is ST6GAL1 – which makes the lack of discussion for other important genes (e.g., FUT2) more noticeable.

One limitation given was the lack of differentiation between vaccine types? Why was this not done? The data exists in the UK Biobank – conducting a secondary analysis of the top hits from the entire GWAS in subcohorts with each vaccine should be fairly easy to do.

Line 145 -Reference 11 is cited as discrepant from reference 10, in that the findings may not be generalizable to the wider community. How so? Do the findings of this pre-print differ from the current manuscript? Are the reported findings supportive of wider generalizability or not?

Figure 5 should come earlier in the manuscript, I recommend that it be Figure 2, provided right after or alongside Figure 1.

Supplementary Table 1 needs a legend. None of the variables on the bottom half are explained.

We thank the Reviewers for their revision and comments provided.
Pages and lines throughout the response refer to the manuscript without track changes.

Reviewer #1 (Remarks to the Author):

The publication of these data and results would be timely and of considerable interest. What the study suffers in messiness (or more formally "noise") it makes up for with sample size.

Some of the main limitations of the study are acknowledged and discussed, especially the fact that most of the antibody response data are non-quantitative and the result of self-administered home tests (if I understand correctly). The sensitivity and specificity of these tests are probably additional concerns (worsened by variable comprehension and care of the self-administered instructions), and the biggest limitation of all may be the variable time between receipt of a vaccine dose and the administration of the test. Presumably a large number of tests were performed within mere days of the vaccination, which may contribute to high apparent non-response? For the second dose in particular (if not also the first dose), the response rate strikes me as low, but maybe the authors can compare these with more careful and quantitative vaccine trial data.

Thank you for the insightful comment. We agree that the antibody response is a broad biological concept that may hardly be represented by a single non-quantitative/binary phenotype. We have thus modified the use of "antibody response" to "seroconversion" throughout the manuscript, which should be more scientifically accurate in this context.

According to the official documents, the antibody tests used in our study are designed for research purposes, and the accuracy is high enough (sensitivity of 98.4% for IgG and 95.2% for IgM, and specificity of 98.8% for IgG and 96.0% for IgM). We hope this will ease your concern on this point.

Lastly, we acknowledge that the time between receipt of a vaccine dose and antibody test is not fixed for everyone in our cohort. We agree with the reviewer that tests performed within mere days of vaccination can lead to apparent non-response, as well as we also agree with another reviewer's comment suggesting that antibody positivity may be too low after 12 weeks. Based on our previous research, we have evidence that antibody positivity may be too low after one week of the first vaccination see Figure 2.B of our recently published article: <https://www.nature.com/articles/s41467-024-48339-5/figures/2>):

[redacted]

Fig 2B. The proportion of individuals with a positive antibody response, stratified by the HLA-DQB1*06 status in the CS-first-dose cohort. Individuals with one or two DQB1*06 alleles are classified as carriers, whereas those with zero DQB1*06 alleles are non-carriers

Hence, we have restricted the analysis to people that performed the antibody test from 1 to 8 weeks (8 to 56 days) after COVID-19 vaccination. Results have been updated accordingly. Additionally, we have provided additional information (here and in supplementary table 1) on the time distribution to facilitate the understanding of our data:

Supplementary figure 1. Histogram of the number of days between vaccination date and antibody test date. The red line indicates the mean value (38.4 for the one dose cohort, 24.7 for the two doses cohort).

The average seropositivity rate (~30% after the 1st dose and 70% after the 2nd dose) is relatively low compared to prior trial data. The potential reason could be multifactorial and likely includes (1) the much older population than previous trial participants, and (2) the fact that we are relying in one single test, which might not capture all individuals that seroconvert. As we acknowledge that this last point was not discussed in the original manuscript, we have added this issue into the discussion section (lines 192-194):

“Second, while seropositivity rates in our study are lower than those reported in clinical trials, this can be attributed to the relatively high age of our participants and the reliance on a single serological test result within a time window, which may not capture all individuals that seroconvert.”

They acknowledge that vaccine type is not accounted for and may be a significant source of heterogeneity. There is a credible effort at distinguishing between positive antibody test results from prior infection vs vaccine alone, although I didn't see these data presented and actually used fully.

Thank you very much for your comment. We have clarified this in the methods section (lines 274-278):

“Responders were defined as participants with a positive serological test (from the COVID-19 self-test antibody seroprevalence study) within 8 to 56 days after latest dose of vaccination. However, individuals that after a seropositivity result had evidence of prior COVID-19 infection (positive result from the COVID-19 infection seroprevalence study), were excluded from the cohort. Non-responders were participants with a negative serological test within 8 to 56 days post-vaccination (see Figure 1A).”

We have also improved the label in Figure 2A, which shows the flow chart on how the seroconversion cohorts were built.

The statistical power for the severity tests and perhaps even susceptibility are much reduced and is probably an unacknowledged reason for the diminishing number of associations, as well as the crudeness of the definition of severity and the likelihood that large numbers of mild cases went unreported and unrecognized by the subjects. The fact that HLA class II alone seems to be driving

antibody response (to a surface antigen) seems to be a common hallmark of vaccine response studies of this sort. So it's a nice result and adds more data to the edifice on this subject. In many ways Figure 2C may be the most interesting, since this turns up additional associations which probably relate to the antigenically more complicated response to a full viral infection. One would predict these associations would more closely resemble the COVID susceptibility/severity studies prior to or independent vaccination and I think these results could be more explicitly and fully compared with these prior GWAS's.

Thank you for highlighting this. We have now included the statistical power in our limitations section (line 205-207):

“Lastly, sample size in genome-wide association studies plays a key role in detecting signals. Some of our analyses, especially the breakthrough severity phenotype, might be underpowered, making it more challenging to detect significant variants.”

We have also acknowledged that unreported mild cases may also influence our analysis (lines 195-198):

“Thirdly, we used linked routinely collected data to ascertain status of COVID-19 infections. This may lead to the outcome misclassification given that asymptomatic or mild infection cases are common particularly among the vaccinated individuals who may not seek for testing and thus falsely classified as controls.”

We have also added more detailed information about the closely resemblance between our results and previous GWAS conducted in COVID-19 susceptibility and severity studies (lines 161-188).

The question of how fully a single (surface-)antigen vaccine can protect vs. more antigenically complicated vaccination strategies seems a worthwhile avenue to explore in light of results like these. I suppose the use of REGENIE for the association analysis is fine, although I was not familiar with it previously and it's not clear it offers any tangible advantages here over the traditional GWAS analytical approach. It seems to have been developed for the simultaneous analysis of multiple independent traits on large biobank style GWAS data sets, which probably isn't much of an issue here. Also, the two-stage design of directly typed followed by imputed SNPs, while probably fine, didn't seem to add very much. Were any associations visible only with imputed variants? If not, they seem to have added nothing here, although presumably they could aid with fine-mapping or identification of more directly functional variants. But none of that was shown or discussed here.

Thank you very much for your comment.

REGENIE is a machine learning method for fitting a whole-genome regression model. While its main strength relies on enabling simultaneous analysis of multiple independent traits, there are also other key features that were particularly advantageous for our analyses.

Firstly, REGENIE uses linear mixed models¹, which allows for better control of confounding due to the complex population structure, including relatedness and population stratification. This allowed us to include participants that were related to each other and hence, increase the sample size of the analyses.

Secondly, REGENIE supports the Firth-regression approach to address case-control imbalances. This method was particularly useful for the breakthrough severity analysis, where the ratio of severe to mild infection was 1:18.

Additionally, REGENIE is well-implemented on the Cloudy-based computation platform (UK Biobank RAP <https://ukbiobank.dnanexus.com/landing>), which can be easily shared with other researchers of interest to replicate or verify. Hence, we believe it is a sensible method to use in our study.

Also, the validation of the results in non-European descent individuals is worthwhile but potentially problematic. It's unclear who these people are, and whether they are simply all non-whites from throughout the world or a single large sub-population within Britain (e.g., South Asian or African/Caribbean). If it's simply all non-whites that is potentially problematic. And of course, if these data are meaningful at all, the fairly large number of partially validated results in Fig. 3 demand more full exploration. And the two that show opposite associations (or total non-validation) for the 2-dose vaccine response may say something important. It may also indict the decision to consider one vs. two dose responses. Incidentally, is 1st dose response data also available for the 2-dose response group, since it appears individuals were only included in one of the two analyses. If 1st dose data are available it might be worth including them in a 1st dose analysis, even if this complicates the accounting for non-independent data in the 2nd dose study.

Thank you for highlighting this. With the aim to provide more information about the non-European descent individuals, we have added Supplementary Table 3, which provides the population characteristics of the validation cohort. In this table, the ethnic background proportions are specified.

Regarding the results, all the genomic loci identified were validated or partially validated. Only one SNP associated with two-dose seroconversion (rs9275766) was not validated (OR = 1.18 in White cohort, $OR_v=1.00$ in the validation cohort).

We agree with the reviewer that including 1st dose response data for the 2-dose response group would be valuable. Unfortunately, we do not have all necessary data to conduct this analysis, as antibody response test were only performed once for each individual in the cross-sectional survey.

Reviewer #2 (Remarks to the Author):

This manuscript documents a GWAS in a large cohort (200,000+) of individuals who received COVID-19 vaccination. The outcomes of interest including serologic response to vaccination, breakthrough infection, and development severe disease. The manuscript is well-written, covers an important topic, has results that nicely complement the multiple publications looking at genetic influences on COVID-19 disease susceptibility and/or severity. The data support the conclusions drawn. The methods are clearly documented and reflect standard approaches typically used.

There are a number of items that the authors should address to strengthen the manuscript and provide increased clarity for readers.

The title and abstract mention antibody response, leading to the assumption that vaccine immunogenicity and magnitude of response will be evaluated. This is not correct, the primary analysis outcome is actually lateral flow assay results that provide a yes/no indicator of seroconversion. In the current manuscript this is not clarified until the limitations paragraph in the Discussion (line 161). This distinction is important and needs to be made clear early on in the manuscript. The authors could change the title from 'antibody response' to 'seroconversion' and/or spell out the outcome in the abstract (line 38-39).

Thank you very much for highlighting this.

We have made this revision as suggested throughout the manuscript.

Following up on this – the authors don't have the actual results of the assay, but are relying on self-reported results. This should be more clear in the limitations paragraph in the Discussion.

Thank you for your comment.

We have further clarified our study features in the discussion (lines 188-192):

"First, we only accounted for a binary seroconversion response, which may have reduced GWAS statistical power compared to quantitative antibody levels. However, this is due to the use of a rapid validated lateral flow device, which allowed for the collection of the largest cohort of individuals self-tested for vaccine immune/antibody response to date, including over 200,000 participants."

Regarding your comment, "the authors don't have the actual results of the assay, but are relying on self-reported results," we apologize for not fully understanding it. It would be greatly appreciated if you could expand a bit more what you mean by saying "actual results vs self-reported results"?

Also, the antibody test was taken within 84 days of vaccination. Why this time period? 84 days provides plenty of time for antibody waning. This needs to be addressed – the analysis may not separate out responders from true non-responders, the latter group may include low responders whose titers waned below the level of detection by the time they tested.

Thank you very much for your comment.

We agree with the reviewer that 84 days window may be too long for antibody waning, as well as with another reviewer comment suggesting that performing tests after mere days of vaccination may be too short for testable antibody response. Hence, we have repeated the analysis restricting to people that performed the antibody test from 1 to 8 weeks (8 to 56 days) after COVID-19 vaccination. Reassuringly, our primary findings remain robust, and all results have been updated accordingly throughout.

Another issue is the repeated use of 'cases' and 'controls'. Each of the 4 analyses use these terms but define them differently. It would be more clear if the authors could use different terminology for each analysis. For example: one-dose responders vs non-responders; two-dose responders vs non-responders; breakthrough infection vs no infection; mild disease vs severe disease.

Thank you very much for highlighting this. We have rectified the terms in the manuscript.

No details are provided for the validation cohort of non-European ancestry. Nor was there any discussion of how the different genetic makeup affects the validation results. What about genetic variants that operate in Caucasians but not other racial groups (or vice versa)?

Thank you very much for your comment. We have now added more details about the validation cohort of non-European ancestry (see Supplementary Table 3), as well as discussing more extensively the results from the validation analysis in the discussion section.

Additionally, we have highlighted the issue about variants operating in certain ethnic groups in the limitations section (lines 200-205):

“Sixth, external replication of the genetic association in other populations is currently impossible in our study due to the lack of similar linked data elsewhere. To mitigate this, we conducted an internal validation by splitting our cohort into those with European and non-European ancestry. Although we did not find apparent evidence of ethnic-specific genetic effects, further studies with larger and more diverse population remain needed to guarantee the findings.”

The discussion section lacks details about the genes associated with each outcome. It does provide information from other studies that have found the same or similar regions, but there is no information given about what each gene does or how it might be related to antibody response, breakthrough infection, or disease severity. The one exception is ST6GAL1 – which makes the lack of discussion for other important genes (e.g., FUT2) more noticeable.

Thank you very much for your comment. We have extended our discussion and provided more details on potential mechanisms from which these genes might impact our outcomes. Please, refer to lines 160-187).

One limitation given was the lack of differentiation between vaccine types? Why was this not done? The data exists in the UK Biobank – conducting a secondary analysis of the top hits from the entire GWAS in subcohorts with each vaccine should be fairly easy to do.

Thank you for highlighting this.

We do not have full information on the type of COVID-19 vaccines received for the whole cohort.

Line 145 -Reference 11 is cited as discrepant from reference 10, in that the findings may not be generalizable to the wider community. How so? Do the findings of this pre-print differ from the current manuscript? Are the reported findings supportive of wider generalizability or not?

Thank you very much. We have clarified our explanation, where we refer that the dimension of the effect might not be generalisable. Please, refer to lines 147-154:

“Human leukocyte antigens (HLA) have been previously acknowledged as the most influential genetic factors for seroconversion to various vaccines²⁻⁴. Mentzer et al.⁵ recently investigated genetic variations associated with 28-day COVID-19 one-dose vaccination antibodies in a cohort of 1,076

participants enrolled in ChAdOx1 nCov-19 vaccine efficacy trials. Their study pinpointed the association between HLA locus with IgG antibody levels and risk of breakthrough infection. Although our study⁶ showed that the effect sizes of the later finding may not be generalisable to the wider community population, the study showed that HLA region plays a key role in antibody response. We provide evidence that HLA region also plays a key role in the COVID-19 vaccine-induced seroconversion.”

Figure 5 should come earlier in the manuscript, I recommend that it be Figure 2, provided right after or alongside Figure 1.

Thank you very much for your comment. We have corrected in the results section.

Supplementary Table 1 needs a legend. None of the variables on the bottom half are explained.

Thank you very much for your comment. We have added the legend, removed the variables on the bottom half and keep the general Index of Multiple Deprivation value.

References

1. Mbatchou, J. *et al.* Computationally efficient whole-genome regression for quantitative and binary traits. *Nat Genet* **53**, (2021).
2. Pulendran, B. Immunology taught by vaccines. *Science (1979)* **366**, 1074–1075 (2019).
3. Pulendran, B. & Davis, M. M. The science and medicine of human immunology. *Science (1979)* **369**, eaay4014 (2020).
4. Dendrou, C. A., Petersen, J., Rossjohn, J. & Fugger, L. HLA variation and disease. *Nat Rev Immunol* **18**, 325–339 (2018).
5. Mentzer, A. J. *et al.* Human leukocyte antigen alleles associate with COVID-19 vaccine immunogenicity and risk of breakthrough infection. *Nat Med* **29**, (2023).
6. Xie, J. *et al.* Relationship between HLA genetic variations, COVID-19 vaccine antibody response, and risk of breakthrough outcomes. *Nat Commun* **15**, 4031 (2024).

REVIEWERS' COMMENTS

Reviewer #1 (Remarks to the Author):

My concerns (and those of the other reviewer) were addressed to my satisfaction. The size of the study and relevance of the topic merit publication.

Reviewer #2 (Remarks to the Author):

The authors have responded satisfactorily to all of my comments. The additions to the text and new data/figures significantly enhance the manuscript. I recommend publication.

Reviewer #2 (Remarks on code availability):

I do not have the coding expertise to evaluate the code provided.